# Extra-Gastric Manifestations of *Helicobacter pylori* Infection

**DOI:** 10.3390/jcm9123887

**Published:** 2020-11-30

**Authors:** Antonietta G. Gravina, Kateryna Priadko, Paola Ciamarra, Lucia Granata, Angela Facchiano, Agnese Miranda, Marcello Dallio, Alessandro Federico, Marco Romano

**Affiliations:** Hepatogastroenterology Division, Department of Precision Medicine, University of Campania Luigi Vanvitelli, via Pansini 5, 80131 Naples, Italy; kateryna.priadko@unicampania.it (K.P.); pciamarra@hotmail.com (P.C.); luciagranata91@gmail.com (L.G.); angela.facchiano@gmail.com (A.F.); mirandaagnese@gmail.com (A.M.); marcello.dallio@gmail.com (M.D.); alessandro.federico@unicampania.it (A.F.)

**Keywords:** extra-gastric manifestations, *H. pylori*, vitamin B12 deficiency anemia, iron deficiency anemia, ophthalmic manifestations, dermatologic manifestations, IBD, eosinophilic esophagitis, Parkinson’s disease, diabetes mellitus, allergy

## Abstract

*Helicobacter Pylori* (*H. pylori*) is a Gram-negative flagellated microorganism that has been extensively studied since its first isolation due to its widespread diffusion and association with numerous diseases. While the bacterium is proved to be a causative factor for a number of gastric diseases such as gastritis, gastric adenocarcinoma, and MALT-lymphoma, its role at other gastrointestinal levels and in other systems is being thoroughly studied. In this article, we reviewed the latest published clinical and laboratory studies that investigated associations of *H. pylori* with hematologic diseases such as Vitamin B12- and iron-deficiency anemia, primary immune thrombocytopenia, and with a number of dermatologic and ophthalmic diseases. In addition, the putative role of the bacterium in inflammatory bowel diseases, esophageal disorders, metabolic, diseases, neurologic diseases and allergy were outlined.

## 1. Introduction

*Helicobacter pylori* (*H. pylori*) is a spiral-shaped flagellated Gram-negative bacillus discovered in 1982. The route of transmission is fecal–oral or oral–oral. *H. pylori* infection is of a great social significance, since its prevalence is estimated to be a half of the world’s population, the highest in developing countries [1,2,3,4]. The colonization of the stomach in most cases happens in early childhood and spreads within families. Current evidence does not support massive transmission in adulthood or among sexual partners. However, a recent study raised the possibility that transmission of the infection may take place between sexual partners, in particular if the couple has been living together for a long period of time and if one of the partners has frequent episodes of gastric regurgitation [5].

The pathogenic activity of *H. pylori* is contributed to by bacterial virulence factors such as CagA, VacA, BabA2, and urease [6]. *H. pylori* infection may undertake different pathogenic and clinical course and is associated with chronic gastritis, peptic ulcer disease, distal gastric adenocarcinoma, and MALT-lymphoma [6,7]. In the vast majority of patients (i.e., approximately 85%), *H. pylori* infection causes a mild mixed gastritis with no alterations of gastric acid homeostasis and is asymptomatic with no relevant clinical outcomes. The variety of clinical conditions, phenotypes, and complications associated with *H. pylori* infection largely depends on the set of socioeconomic factors, bacterial virulence factors, and genetic predisposition of the host (e.g., *IL1B* gene cluster and tumor necrosis factor-α (*TNFα*) gene polymorphism) [8]. While the majority of infected individuals suffer from mild antrum/corpus gastritis, 1–2% of infected subjects will develop multifocal atrophic gastritis and, at some point, gastric adenocarcinoma [8]. Inhibition of apoptosis, DNA mutations, cell proliferation, and increased growth factor or cyclooxygenase-2 expression are the carcinogenic pathways possibly involved in the development of gastric cancer following *H. pylori* infection [9,10]. Based on this, *H. pylori* must be eradicated when found [11]. The first-line eradication regimens, especially in areas of high clarithromycin resistance, are bismuth quadruple or non-bismuth quadruple (i.e., concomitant) therapy for 14 days [12,13].

While the association of the *H. pylori* infection with gastric pathologic conditions is well established, the effects of the bacterium on the whole host organism are still being questioned, as it is generally known that certain pathogens, remaining local, may exert systemic pathological effects [13]. The greatest concern about *H. pylori* is not just about its causal role in extra-gastric diseases but also about its capability of disease phenotype modification. Current review focuses on hematologic, ocular, and dermatologic manifestations of *H. pylori* infection. Additionally, its putative role in inflammatory bowel diseases (IBD) is evaluated (Figure 1).

## 2. Hematological Diseases

### 2.1. Vitamin B12 Deficiency Anemia

Vitamin B12 (Vit. B12) deficiency anemia is a pathological condition due to the lack of intake or absorption of vitamin B12, which serves as a coenzyme in many metabolic reactions, contributing to DNA synthesis. In a chronic course of Vit. B12 deficiency, a megaloblastic anemia, neuropathy, and other pathological conditions (i.e., pernicious anemia) may occur [14,15]. Lahner et al., in a systematic review including 2454 subjects, demonstrated a relationship between low serum levels of Vit. B12 and *H. pylori* infection [16]. Additionally, low serum levels of Vit. B12 and increased serum levels of homocysteine, a metabolic product of Vit. B12, have been described in *H. pylori*-infected subjects. In further support of a relationship between *H. pylori* infection and Vit. B12 deficiency, the serum levels of Vit. B12 and of homocysteine returned to normal following successful *H. pylori* eradication [13]. Decreased absorption of Vit. B12 in case of *H. pylori* infection might be due to intrinsic factor deficiency, normally produced by gastric parietal cells, that are damaged in case of *H. pylori*-related corpus-predominant gastritis [17]. Moreover, *H. pylori* infection may also be associated with autoimmune atrophic gastritis, which is characterized by a diffuse corpus atrophic gastritis and presence of antibodies toward intrinsic factor. Because of this scientific evidence, all patients of unexplained megaloblastic anemia associated with Vit. B12 deficiency should undergo a diagnostic test for *H. pylori* infection and, if positive, should be eradicated of the infection.

### 2.2. Iron-Deficiency Anemia

Iron-deficiency anemia (IDA) is one of a few extra-gastric conditions in which *H. pylori* eradication is strongly recommended by Maastricht V/Florence guidelines as well as by guidelines on IDA [13]. In fact, several metanalyses clearly showed a strong association between *H. pylori* infection and IDA [18,19]. One of the major causes of IDA during *H. pylori* infection is blood loss due to bleeding (i.e., ulcer bleeding) in infected patients. However, *H. pylori*-associated IDA may also occur in patients with intact mucosa, thus leading to further investigations in order to better define the pathophysiological mechanisms underlying such association. One of the pathways through which *H. pylori* infection may lead to iron deficiency is by affecting iron absorption by enterocytes and its release from macrophages via hepcidin level upregulation [20]. Yokota et al. demonstrated more frequent gene polymorphisms within the neutrophil-activating protein of *H. pylori* strains in patients with *H. pylori*-associated IDA than in those without IDA [21]. Senkovich et al. demonstrated that *H. pylori* is able to acquire iron from host glycoproteins such as transferrin and lactoferrin and that this contributes to the persistence of the infection. Furthermore, an increased intragastric pH due to *H. pylori*-associated corpus-predominant chronic gastritis may impair the transformation of dietary Fe^3+^ to Fe^2+^, thus decreasing the duodenal absorption of iron [22]. Additionally, *H. pylori* may up-regulate the expression of TNFα, which is a pro-inflammatory cytokine that may cause IDA. In support of a role for *H. pylori* in inducing IDA, it has been demonstrated that eradication of the infection leads to a reversal of IDA in up to 75% of patients [18]. The relationship between *H. pylori* and IDA is also relevant as to *H. pylori*-related gastric carcinogenesis [23]. It has in fact been shown that iron deficiency may favor further colonization of the stomach by the bacterium and also facilitates the assembly of the Type 4 Secretion System through which *H. pylori* injects CagA into gastric epithelial cells [24]. CagA is well known to exert proinflammatory effects and to stimulate, following its phosphorylation, intracellular pathways, which stimulates proliferation and decrease apoptosis, thus favoring the development of gastric cancer [25].

### 2.3. Primary Immune Thrombocytopenia (Formerly Known as Idiopathic Thrombocytopenic Purpura)

Primary immune thrombocytopenia (PIT) is an autoimmune disorder, which has been shown to be linked to *H. pylori* infection. The potential pathogenic role of *H. pylori* is suggested by many studies showing that successful eradication of *H. pylori* infection leads to a raise of platelet count ranging from 26 to 100% depending on ethnic and racial identities/origins [26,27]. The pathogenic link between *H. pylori* infection and PIT might be the molecular mimicry between platelet surface glycoproteins and amino acid sequences of *H. pylori* virulence factors, in particular the *H. pylori* lipopolysaccharide (LPS). Additionally, *H. pylori* up-regulates Fcγ receptor expression thus increasing the phagocytic capacity and down-regulates inhibitory receptors FcyRIIB that in turn enhances monocyte activity and autoreactivity with B and T lymphocytes [28,29,30,31]. As a result, B-lymphocytes produce autoantibodies to circulating platelets. Some studies have indicated that polymorphism in *IL-β*, the *IL-β* (-511) T allele, represents a predisposing factor to PIT development in *H. pylori* positive subjects [30,31]. In partial support of this hypothesis, eradication of *H. pylori* in PIT patients results in an upregulation of FcγRIIB expression in monocytes and disorganization of antigen presentation that leads to unrecognition of platelet antigens [28,29,30,31]. Maastricht V/Florence guidelines and guidelines on management of PIT suggest considering testing for and eradication of *H. pylori* in subjects with PIT [13].

## 3. Ophthalmic Diseases

### 3.1. Open-Angle Glaucoma

Open-angle glaucoma (OAG), a condition that may lead to optic nerve damage, has been evaluated as a putative *H. pylori* extra gastric feature. Even though studies evaluating the prevalence of *H. pylori* in patients with OAG present controversial results, in a meta-analysis evaluating 10 studies, performed by Zeng et al., a statistically significant correlation was found. On average, the prevalence of infection was twice as high than in control group. In further support of a causative role of *H. pylori* in this condition, patients with OAG showed a normalization of intraocular pressure and mean visual field parameters after *H. pylori* eradication [32].

### 3.2. Central Serous Chorioretinopathy

Central serous chorioretinopathy (CRSC) is a condition characterized by the detachment of serous layer of sensory retina on its posterior side due to the liquid leakage and its accumulation under the macula. Diagnosis is through the use of fluorescein and indocyanine green angiography [33]. Main clinically relevant complaints of affected patients include insufficient vision acuity, micropsia, metamorphopsia and dyschromatopsia [33]. Possible outcomes of CSRC include photoreceptor loss, atrophy, and shortening of the photoreceptor outer segment [33]. Even though the spontaneous resolution is noted relatively often within 4 months after the initiation of the disease, not all the patients become asymptomatic. Even after fluid reabsorption, patients can present such symptoms as blurred and/or distorted vision. [34]. The possible role of *H. pylori* in the pathogenesis of CSRC has been evaluated in a number of studies. In a retrospective observational case series study, the prevalence of *H. pylori* has been assessed in patients with CSRC and in a control group, represented by subjects with ophthalmic disease other than CSRC. The *H. pylori* positive status was confirmed only in case of both HpSA and serology IgG positive results. The study demonstrated an approximately two-fold higher prevalence of *H. pylori* infection in patients with CSRC over control subjects. The odds ratio for CSRC associated with *H. pylori* infection was 4.6 (95% CI 1.28–16.9) [33]. Other studies demonstrated that eradication of the infection leads to a considerable improvement of CSRC and a decrease in recurrence rate of 45% [35,36].

## 4. Dermatologic Diseases

### 4.1. Rosacea

Rosacea is a persistent facial dermatosis with recurrent inflammatory skin elements such as papules, pustulas, oedema, and telangiectasia. Some studies suggest the role of immune disorders, microbial agents, vascular pathology, and UV radiation. Whether *H. pylori* infection plays a role in rosacea is the object of controversy. A study by Gravina et al. reported a significantly increased prevalence of *H. pylori* infection among patients with rosacea compared to controls [37]. Additionally, eradication of the infection was associated to a significant improvement in rosacea cutaneous manifestations [37]. This has not been confirmed in other studies. The possible explanation for this discrepancy could be the difference in methods used to diagnose *H. pylori* infection (i.e., serum IgG anti-*H. pylori* vs. C13 UBT accordingly). The mechanism by which *H. pylori* may cause rosacea could be through an increased tissue and serum level of NO contributing to vasodilation and immunomodulation, which may facilitate erythema and flushing [37].

### 4.2. Psoriasis

Psoriasis is an inflammatory skin and/or joint disease with chronic recurrent course. The prevalence of the disease is 2–3% across the globe and the etiology is multifactorial. Defining a possible pathogenic role of a bacterial agent as *H. pylori* could be of a great significance. Following some studies, the incidence of *H. pylori* in patients with psoriasis showed dramatic superiority (79% with severe psoriasis and 46% with mild) over the incidence in the general population (approximately 30% depending on the region) [38,39]. Furthermore, *H. pylori* infected patients with psoriasis had a statistically significant increase in a Psoriasis Area and Severity Index (PASI) score. The eradication of the infection has been shown to improve the symptoms [40].

### 4.3. Chronic Urticaria

Chronic urticaria (CU) is a skin condition characterized by specific itchy patches defined as wheals. CU is considered to have an autoimmune background in its pathogenesis with the production of autoantibodies inducing histamine release via IgE epitopes or FceRI receptors binding [41]. No increased *H. pylori* prevalence in the CU group was found [42]. A clear correlation between CU and *H. pylori* infection is lacking, and evidence is contradictory. A study found a specific chain of metamorphoses in gastric juice of *H. pylori*-infected subjects, connected with eosinophil cationic protein, being noted exclusively in CU patients [41]. From this perspective, investigations assessing clinical improvement of skin condition after *H. pylori* eradication are crucial. In this respect, Campanati et al. as well as Youshimasu et al. showed a positive outcome of skin lesions after anti-*H. pylori* therapy [42,43].

### 4.4. Alopecia Aerata

Alopecia aerata (AA) is a condition pathogenically related to the production of autoantibodies, which attack hair follicles, thus leading to partial or total hair loss. The prevalence of the disease is 1.7% in the general population [41]. Case reports were published stating an association between *H. pylori* and AA, confirming a stable remission after eradication treatment. At the same time, there are many contradictory data that underline the need to perform new large cohort randomized-controlled studies.

### 4.5. Autoimmune Bullous Disease

Autoimmune bullous disease (AIBD) is a group of autoimmune diseases such as pemphigus, pemphigoid, dermatitis herpetiformis, epidermolysis bullosa acquisita, and linear Ig A disease [41,44] in which autoantibodies attack certain elements of epidermal desmosome and hemidesmosome [41]. Sagi L et al. and Mortazavi H et al. showed an approximately 79% *H. pylori* seropositivity prevalence in an AIBD group vs. healthy controls [45,46], thus suggesting a possible pathogenic role of *H. pylori* in this condition.

## 5. Inflammatory Bowel Diseases

Inflammatory bowel diseases (IBD) form a group of chronic relapsing-remittent diseases affecting gastrointestinal tract (GI). The etiology is complex and includes genetic, immunological, and environmental factors [47,48]. Many studies were performed to assess either the protective or causative role of *H. pylori* in patients with IBD with results showing contradictory data [47,48].

Dozens of *Enterohepatic Helicobacter (EHH)* species have been found to spread within GI tract and induce or sustain intestinal inflammation in rodents. The valuable correlation between *H. pullorum* and *H. canadensis* and Crohn’s disease (CD) in humans was defined by Laharie et al. [49]. In a study by Kaakoush et al., fecal samples and colon biopsies of children with CD and controls were examined to assess prevalence of *Helicobactericae*, and the CD group had approximately a two-fold increase in the prevalence over controls [50]. On the contrary, Ruuska T et al. evaluated the prevalence of *H. pylori* infection in pediatric population with IBD and found 4.8% prevalence in CD and 10.6% prevalence in ulcerative colitis (UC) [51]. This apparent discrepancy between studies might be accounted for by remarkable strain differences observed in clinical isolates.

The majority of studies support the protective potential of the bacterium in IBD. The incidence of IBD in *H. pylori* positive subjects was defined as 32% in UC and 17.7% in CD vs. 52.5% in healthy controls by Song et al. [52]. In a study conducted by Iranian colleagues, evaluating the protective role of *H. pylori* against IBD, a significantly lower prevalence of IBD was found in *H. pylori* positive subjects vs. control group, supporting the data from previous studies [53]. Moreover, the severity of IBD, UC in particular, increased after *H. pylori* eradication [54]. The same information was provided by Shinzaki S et al., who found an improvement of IBD condition only in subjects who did not undergo eradication therapy [55]. The IBD phenotype-modifying capabilities of *H. pylori* were shown in studies demonstrating that fibro-stenotic and fistulizing CD as well as risk of bowel resections and number of relapses were significantly lower in *H. pylori*-infected subjects [48].

The pathophysiologic involvement of *H. pylori* in the setting of IBD might be related to the fact that the bacterial infection triggers a systemic Th1 mediated immunological response, thus stimulating both specific and non-specific immune reactions, which may initiate a chain of immunologic reactions, contributing to IBD. Another possible altering pathway is a direct exposure of intestinal mucosa to urease and cytotoxins [48]. On the other side, a protective role of the bacterium might be contributed to by the influence on a tolerogenic phenotype of dendritic cells and T-regulatory cells bearing immunosuppressive features [47]. *H. pylori* also produces peptides, which exert a protective effect on the inflamed intestinal mucosa. In particular, Hp(2–20) is a 19 amino acid peptide derived from the N-terminal region of *H. pylori* ribosomal protein L1 whose biological effects are mediated through the interaction with N-formyl peptide receptors (FPRs), which are seven transmembrane G protein coupled receptors [56,57,58]. Human FPRs are expressed on the cell membranes of the host immune cells (e.g., neutrophils, monocytes) but also on nonimmune cells such as intestinal adenocarcinoma cells and on the baso-lateralmembrane of human crypt epithelial cells [59,60]. Gravina et al. demonstrated that Hp(2–20) significantly accelerates colonic mucosal healing both at the macroscopic and histological levels in 2,4,6-trinitrobenzenesulfonic acid (TNBS)-induced colitis in rat model and that this effect is associated with a significant reduction in colonic tissue levels of inflammatory mediators (i.e., COX-2 and TNF-α) and a significant upregulation of mediators of damage repair (i.e., TGF-, t-TG, and HB-EGF) [61].

## 6. Esophageal Diseases

*H. pylori* infection has been suggested to exert a protective effect against esophageal diseases. In particular, a lower prevalence of *H. pylori* infection has been demonstrated in patients with gastro-esophageal reflux disease (GERD), Barrett’s esophagus, and esophageal adenocarcinoma compared to controls [62,63]. This has been hypothesized to be contributed to by a decreased acid secretion in *H. pylori*-infected subjects with corpus-predominant gastritis. Whether *H. pylori* eradication leads to an increased prevalence of GERD is controversial [64,65]. *H. pylori* eradication does not seem to exacerbate GERD symptoms, nor does it decrease the efficacy of proton pump inhibitors (PPI) [66]. However, *H. pylori* infection should be eradicated in GERD patients under PPI therapy, because PPI-induced inhibition of gastric secretion increases the risk of development or the severity of atrophic gastritis of the body, which is a premalignant condition [67,68].

Eosinophilic esophagitis (EoE) is a chronic, immune- and allergen-mediated condition characterized by eosinophils infiltration of the esophagus associated to symptoms of esophageal dysfunction (i.e., dysphagia and food impaction). There is no direct evidence linking *H. pylori* infection to EoE, however indirect evidence suggests an inverse relationship between *H. pylori* and EoE (i.e., *H. pylori* is protective against development of EoE). In fact, a recent comprehensive metanalysis by Shah et al. demonstrated that *H. pylori* exposure was associated to a 37% decrease in the odds of EoE [69]. The mechanism whereby *H. pylori* infection might confer a protective effect against the development of EoE may rely in a protective immunoregulatory phenotype of *H. pylori* leading to an attenuation of signaling pathways also involved in allergy with a shifting of the immune response toward cytokines of the Th1 type and a decrease in Th2 polarization [70,71]. A prospective, case-control study by Molina-Infante et al. has, however, reached the conclusion that there is no association whatsoever between *H. pylori* and EoE [72]. Robust prospective trials are needed to properly address this issue, and mechanistic studies are needed to demonstrate whether *H. pylori* itself biologically protects against EoE.

## 7. Metabolic Diseases

*H. pylori* infection has been described to be associated to a number of metabolic disorders, in particular to diabetes mellitus (DM) and metabolic syndrome (MetS). The evidence regarding an association between *H. pylori* and DM is controversial. Hsieh et al. demonstrated that an elevated level of glycated hemoglobin was associated to *H. pylori* in an elderly population [73]. Additionally, Beguè et al. showed that eradication of *H. pylori* led to a better control of glycemia through the assessment of glycated hemoglobin [74]. However, Demir et al. showed that blood glucose levels or glycated hemoglobin levels were not significantly different between *H. pylori*-infected subjects and controls [75]. The putative mechanism whereby *H. pylori* might interfere with glucose metabolism could be through an increase in circulating cytokines, which might interfere with a number of metabolic processes [76]. In partial support of this hypothesis, Wang et al. have demonstrated that patients with DM have higher levels of circulating markers of inflammation compared to controls [77]. Additionally, *H. pylori* may contribute to a disturbance of glucose homeostasis by inducing insulin resistance interfering with the c-Jun/miR-203/SOCS3 pathway [78]. In this context, several studies have demonstrated a higher prevalence of MetS in *H. pylori*-infected subjects than in controls [79]. In particular an increased serum levels of cholesterol and triglycerides has been shown in a Finnish population [80]. 

## 8. Neurological Diseases

*H. pylori* infection has been reported to be associated to a number of neurological disorders, even though most of the studies addressing this issue resulted in controversial results.

In a prospective cohort study in patients with stroke, Chen et al. showed no increased mortality in *H. pylori*-infected subjects compared to controls [81]. On the other hand, a metanalysis by Wang et al. showed that *H. pylori* infection mainly with CagA positive strains represented a risk factor for ischemic stroke [82]. The putative pathogenic mechanism is an increase in mediators of inflammation induced by the infection, which might activate platelets and coagulation [83]. The apparent discrepancy between studies might at least in part be contributed to by differences in the study population.

*H. pylori* infection has also been found associated to Alzheimer’s disease (AD) [84,85]. Additionally, eradication of the infection has led to an improvement of AD-related symptoms [86,87]. The pathogenic link could be an increased prevalence in *H. pylori*-infected subjects of apolipoprotein E (ApoE) polymorphism, which is a risk factor for AD [88]. Additionally, in the cerebrospinal fluid (CSF) of AD patients’ high levels of IgG anti-*H. pylori* have been described [89]. It has been hypothesized that *H. pylori* may access the brain through an oro-nasal pathway, thus leading to neurodegeneration. Alternatively, *H. pylori*-infected monocytes might access the brain through a pro-inflammatory cytokine-mediated disruption of the blood–brain barrier (BBB) [90]. Finally, *H. pylori* might access the brain from the gastrointestinal tract through the enteric nervous system [90].

Parkinson’s disease (PD) has been found to be significantly associated to *H. pylori* infection in a metanalysis involving over 30,000 patients [91]. Additionally, *H. pylori* seems to decrease the bioavailability of L-dopa, the main therapeutic agent for PD, through a damage to the duodenal mucosa where the absorption of L-dopa takes place [92]. A recent metanalysis by Dardiotis et al. demonstrated a higher prevalence of *H. pylori* infection in patients with PD [93]. Moreover, non-infected PD patients and PD patients after *H. pylori* eradication showed lower unified PD rating scores, thus suggesting that the infection may negatively affect the severity of the disease [93]. A number of pathogenic mechanisms have been hypothesized to explain this association. It has been postulated that *H. pylori*-induced production of pro-inflammatory cytokines may cause disruption of the BBB and death of dopaminergic neurons, thus contributing to the development of PD [94]. Moreover, molecular mimicry from *H. pylori* may cause the formation of auto-antibodies against dopaminergic neurons, or *H. pylori* may induce the formation of noxious chemicals in the stomach, which may then be transmitted through the vagal axonal afferent pathway and affect neurons located in the brainstem [93].

Guillain-Barrè syndrome (GBS) is an autoimmune, acute neuropathy, which causes paralysis of limbs with a distal pattern. This disease is often triggered by an infectious disease, and *H. pylori* infection has been described to be associated to GBS [95]. The pathogenic link has been supposed to be a molecular mimicry between *H. pylori* LPS and peripheral nerve gangliosides [95]. Additionally, Chiba et al. demonstrated that the CSF from GBS patients has high levels of IgG anti *H. pylori* VacA [96]. Interestingly, the same authors described a sequence homology between VacA and the human ATPase A subunit, thus implying that anti-VacA antibodies might interact with ion channels of Schwann cells, thus causing a demyelination of motor neurons [96].

## 9. Allergic Diseases

*H. pylori* infection has been shown to be associated with a number of allergic diseases, including asthma. In particular, an inverse relationship between *H. pylori* and allergic disorders, particularly in children and young people, has been described [70,97]. It has been hypothesized that *H. pylori* has evolved so as to shift the adaptive immune response toward tolerance rather than immunity, thus promoting persistence of the infection on the one side and inhibiting auto-aggressive and allergic T-cell responses on the other. VacA toxin seems the most important factor involved in the protective effect exerted by *H. pylori* against allergies by disrupting the Th cell response to regulatory T (Treg) cells [98]. Treg cells are considered the most crucial cellular players in the defense against allergies either in interventional studies in humans or in experimental studies in animals. *H. pylori* CagA positive strains, which express active VacA, not only modulate Treg cell expression but also seem to affect Treg cell migration [99]. The function of Treg cells depends on the production of IL-18 by dendritic cells following exposure to *H. pylori* [100]. Tolerogenic dendritic cells and Treg cells induced by *H. pylori* orchestrate so to prevent adaptive Th1/Th17 driven immune response to the infection and to inhibit allergen specific Th2 responses, thus inducing tolerance and protection against allergies [70]. In the analysis of the association between *H. pylori* and allergy, one should also consider the hygiene hypothesis. *H. pylori* infection is generally associated to poor hygiene conditions and low socio-economic status, which both increase the risk of exposure to other bacteria or antigens, which, in turn, may reduce the risk of allergies. Therefore, epidemiological studies showing a relationship between *H. pylori* and allergies should be interpreted taking into account all possible confounding factors before a firm cause–effect relationship can be drawn.

## 10. Conclusions and Future Directions

*H. pylori* is most commonly associated with gastroduodenal diseases, including peptic ulcer disease and gastric adenocarcinoma, which are the results of an interaction between bacterial virulence factors and host and environmental factors. *H. pylori* infection has also been reported to be linked to several extra-gastric diseases, most of which have been object of this review (Table 1). In many cases the association is based on epidemiological studies, and a cause–effect relationship cannot be established. As a result of this, to date, according to the last Consensus Report on the management of *H. pylori* infection (i.e., Maastricht V/Florence Guidelines [13]) *H. pylori* infection should be sought and, if present, eradicated only in patients with IDA, PIT, and vitamin B12 deficiency. Interventional studies are necessary to establish in a definite manner whether the eradication of *H. pylori* is beneficial to the pleiotropic conditions, which are associated to the infection. Moreover, mechanistic studies are needed to better define whether the association between *H. pylori* infection and extra-gastric manifestations have a well-defined pathogenic pathway so as to possibly identify potential targets of intervention. In particular, it would be of interest to ascertain if it is the inflammatory response generated by the infection or the bacterium itself through its virulence factors to play a role in the development of extra-gastric diseases. Finally, further research is warranted in order to define whether *H. pylori* products, such as Hp(2–20) peptide, might be considered as potential therapeutic agents in specific clinical settings, such as inflammatory bowel diseases.

## Figures and Tables

**Figure 1 jcm-09-03887-f001:**
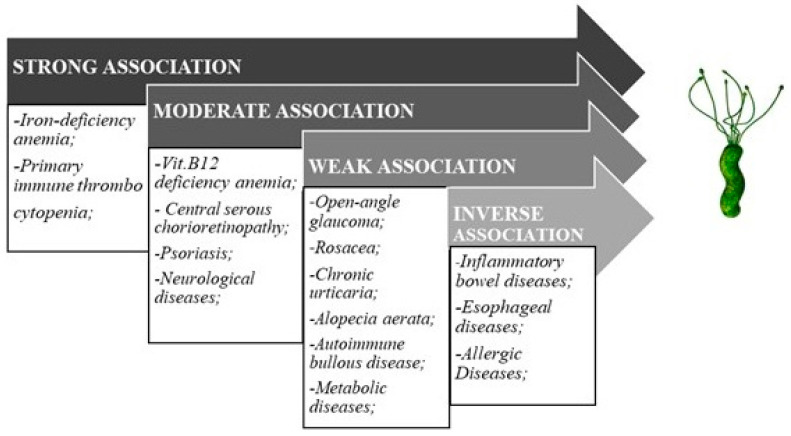
*Helicobacter pylori* and extra-gastric diseases association.

**Table 1 jcm-09-03887-t001:** Association between *H. pylori* infection and extra-gastric diseases.

Diseases	Evidence of Association with *H. pylori* Infection	Key Evidence of Support	References Associated with the Evidence
***Hematological diseases***			
Vitamin B12 deficiency anemia	Moderate evidence for association with *H. pylori*	Decreased absorption of Vit. B12 in case of *H. pylori* infection might be due to intrinsic factor deficiency, normally produced by gastric parietal cells that are damaged in case of *H. pylori*-related corpus-predominant gastritis	Lahner, E. et al. [17] Valdes-Socin, H. et al. [17]
Iron-deficiency anemia	Strong evidence for association with *H. pylori*	- Blood loss due to bleeding (i.e., ulcer bleeding) - Low iron absorption by enterocytes and impaired release from macrophages due to hepcidin upregulation - Increased intragastric pH due to *H. pylori*-associated corpus-predominant chronic gastritis may impair the transformation of dietary Fe^3+^ to Fe^2+^, thus decreasing the duodenal absorption of iron	Malfertheiner, P. et al. [13] Hudak, L. et al. [18] Goddard, A.F. et al. [19] Mendza, E. et al. [20] Yokota, S.I. et al. [21] Senkovich, O. et al. [22] El-Omar, E.M. et al. [23] Noto, J.M.v. et al. [24] Sokolova, O. et al. [25]
Primary immune thrombocytopenia (formerly known as idiopathic thrombocytopenic purpura)	Strong evidence for association with *H. pylori*	Molecular mimicry between platelet surface glycoproteins and amino acid sequences of *H. pylori* virulence factors	Malfertheiner, P. et al. [13] Suvajdzic, N. et al. [26] Jackson, S.C. et al. [27] Campuzano-Maya, G. et al. [28] Asahi, A. et al. [29] Testerman, T.L. et al. [30] Satoh, T. et al. [31]
***Ophthalmic diseases***			
Open-angle glaucoma	Weak evidence for association with *H. pylori*	Unknown	Zeng, J. et al. [32]
Central serous chorioretinopathy	Moderate evidence for association with *H. pylori*	Unknown	Cotticelli, L. et al. [33] Zavoloka, O. et al. [35] Dang, Y. et al. [36]
***Dermatologic diseases***			
Rosacea	Weak evidence for association with *H. pylori*	Increased tissue and serum level of nitric oxide contributing to vasodilation and immune-modulation, which may facilitate erythema and flushing	Gravina, A.G. et al. [37]
Psoriasis	Moderate evidence for association with *H. pylori*	Unknown	Mesquita, P.M. et al. [38] Mesquita, P.M. et al. [39] Onsun, N. et al. [40]
Chronic urticaria	Weak evidence for association with *H. pylori*	Unknown	Campanati, A. et al. [42] Yoshimasu, T. et al. [43]
Alopecia aerata	Weak evidence for association with *H. pylori*	Unknown	Magen, E. et al. [41]
Autoimmune bullous disease	Weak evidence for association with *H. pylori*	Unknown	Sagi, L. et al. [45] Mortazavi, H. et al. [46]
***Inflammatory bowel diseases***			
Ulcerative colitis and Crohn’s disease	Moderate evidence against association with *H. pylori*	Tolerogenic phenotype of dendritic cells and Treg cells with immune-suppressive features	Yang, Y. et al. [47] Papamichael, K. et al. [48] Kaakoush, N.O. et al. [50] Ruuska, T. et al. [51] Song, M.J. et al. [52] Sayar, R. et al. [53] Jin, X. et al. [54] Shinzaki, S. et al. [55]
***Esophageal diseases***			
Gastro-esophageal reflux disease (GERD)	Moderate evidence for inverse relationship with *H. pylori*	Decreased acid secretion in *H. pylori*-infected subjects with corpus-predominant gastritis	Fischbach, L.A. et al. [62] Lee, Y.C. et al. [64] Yaghoobi, M. et al. [65]
Eosinophilic esophagitis (EoE)	Moderate evidence for inverse relationship with *H. pylori*	Protective immune-regulatory phenotype of *H. pylori* leading to an attenuation of signaling pathways also involved in allergy with a shifting of the immune response toward cytokines of the Th1 type and a decrease in Th2 polarization	Arnold, I.C. et al. [70] Arnold, I.C. et al. [71] Molina-Infante, J. et al. [72]
***Metabolic diseases***			
Diabetes mellitus and increased serum levels of cholesterol and triglycerides	Weak evidence for association with *H. pylori*	Increase in circulating cytokines, which might interfere with a number of metabolic processes	Hsieh, M.C. et al. [73] Bèguè, R.E. et al. [74] Demir, M. et al. [75] Calle, M.C. et al. [76] Wang, X. et al. [77] Zhou, X. et al. [78] Chen, T.P. et al. [79] Niemela, S. et al. [80]
***Neurological diseases***			
Ischemic stroke	Weak evidence for association with *H. pylori*	Increase in mediators of inflammation induced by the infection, which might activate platelets and coagulation	Chen, Y. et al. [81] Wang, Z.W. et al. [82] Alvarez-Arellano, L. et al. [83]
Alzheimer’s disease	Weak evidence for association with *H. pylori*	Increased prevalence in *H. pylori*-infected subjects of apolipoprotein E (ApoE) polymorphism, which is a risk factor for Alzheimer’s disease	Huang, W.S. et al. [84] Roubaud, B.C. et al. [85] Kountouras, J. et al. [86] Kountouras, J. et al. [87] Kountouras, J. et al. [88] Kountouras, J. et al. [89]
Parkinson’s disease	Moderate evidence for association with *H. pylori*	Production of pro-inflammatory cytokines may cause disruption of the blood–brain barrier and death of dopaminergic neurons	Shen, X. et al. [91] Dardiotis, E. et al. [83] Dobbs, R.J. et al. [94]
Guillain-Barrè syndrome	Moderate evidence for association with *H. pylori*	Molecular mimicry between *H. pylori Lipopolysaccharide* (LPS) and peripheral nerve gangliosides	Kountouras, J. et al. [95] Chiba, S. et al. [96]
***Allergic Diseases***	Moderate evidence against association with *H. pylori*	Shifting of the adaptive immune response toward tolerance rather than immunity, thus promoting persistence of the infection on the one side and inhibiting auto-aggressive and allergic T-cell responses on the other	Blaser, M.J. et al. [97] Oertli, M. et al. [98] Cook, K.W. et al. [99] Oertli, M. et al [100]

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
