# Peer review of "Extra-Gastric Manifestations of Helicobacter pylori Infection"

_jcm, 2020, doi:10.3390/jcm9123887_

Round 1
Reviewer 1 Report
Summary
The authors nicely reviewed evidence in the literature for and against the association of H. pylori with extra-gastric diseases such as inflammatory bowel disease, ocular diseases, primary immune thrombocytopenia, dermatological diseases, and vitamin B12/iron deficiency anemia. The article does a nice job of summarizing key studies with good balance of opposing studies, especially the IBD section.
Major suggested changes
- There is a serious problem with the references throughout the manuscript. There are two ref. #1 in the bibliography and it appears the numbering is off by 1 throughout the manuscript (eg. Line 100-104, Senkovich is ref 21 not 22). The authors are urged to closely examine the references and make sure they are correct throughout. Additionally, bacteria genus/species names and gene names (not protein names) need to be italicized in the references.
- I strongly recommend creating a summary table, with each disease in first column, 2nd column simple statement of whether there is weak, moderate or strong evidence of association (for or against) with H. pylori, 3rd column a little more detail on the specific key evidence of support, 4th column references associated with the evidence. Then modify Figure 1 slightly with arrows between each circle and connecting it to H. pylori, with an arrow thickness corresponding to weak (thin arrow), moderate (thicker arrow), and strong (thickest arrow) association. This way both a table and the figure will have the key take home messages for readers of the article.
- The authors seemingly ignored the growing body literature on the association of H. pylori and Parkinson’s disease. Could they add a section on this area?
Minor suggested changes
- Line 84 & 87. Change “to” to “with”
- Line 86. Change “this” to “these”
- Lines 118-120. Expand slightly- what virulence factors associated with this- is it the well-known Lewis antigen mimicry in the H. pylori LPS?
- Line 126. Change “PTI” to “PIT”
- Line 150. Change “n” to “in”
- Lines 208-248. Perhaps speculate that the reasons for the conflicting data on IBD association with H. pylori is due to the remarkable strain diversity observed in clinical isolates?
- Line 253. Change “despit” to “despite”
- Line 258-259. Sentence unclear. Perhaps change to: “….if H. pylori eradication can lead to improvement in a number of extra-gastric diseases.”
Author Response
We thank the reviewer for her/his comments
- There is a serious problem with the references throughout the manuscript. There are two ref. #1 in the bibliography and it appears the numbering is off by 1 throughout the manuscript (eg. Line 100-104, Senkovich is ref 21 not 22). The authors are urged to closely examine the references and make sure they are correct throughout. Additionally, bacteria genus/species names and gene names (not protein names) need to be italicized in the references.
Response: We thank the reviewer for his comment and apologize for this mistake that has now been taken care of. Also bacteria genus/species are in italics in the reference section
- I strongly recommend creating a summary table, with each disease in first column, 2nd column simple statement of whether there is weak, moderate or strong evidence of association (for or against) with H. pylori, 3rd column a little more detail on the specific key evidence of support, 4th column references associated with the evidence. Then modify Figure 1 slightly with arrows between each circle and connecting it to H. pylori, with an arrow thickness corresponding to weak (thin arrow), moderate (thicker arrow), and strong (thickest arrow) association. This way both a table and the figure will have the key take home messages for readers of the article.
Response: Per the Reviewer’s request we have created a table according to the reviewer’s suggestions. We also have modified Figure 1 highlighting the strength of the association between H. pylori and the related extragastric-manifestations
- The authors seemingly ignored the growing body literature on the association of H. pylori and Parkinson’s disease. Could they add a section on this area?
Response: because of the reviewer’s concern we have now introduced a new section on Neurological Disease and H. pylori in which, among others neurological disorders, we also discuss the evidence which relates H. pylori infection to Parkinson’s disease
Minor suggested changes
- Line 84 & 87. Change “to” to “with”
Response: done
- Line 86. Change “this” to “these”
Response: done
- Lines 118-120. Expand slightly- what virulence factors associated with this- is it the well-known Lewis antigen mimicry in the H. pylori LPS?
Response: we added a sentence including the referee’s suggestion
- Line 126. Change “PTI” to “PIT”
Response: done
- Line 150. Change “n” to “in”
Response: done
- Lines 208-248. Perhaps speculate that the reasons for the conflicting data on IBD association with H. pylori is due to the remarkable strain diversity observed in clinical isolates?
Response: we agree with the reviewer’s comment and added a sentence regarding the role of the strain diversity of clinical isolates as to the discrepancy between data on IBD
- Line 253. Change “despit” to “despite”
Response: done
- Line 258-259. Sentence unclear. Perhaps change to: “….if H. pylori eradication can lead to improvement in a number of extra-gastric diseases.”
Response: done

Reviewer 2 Report
Reviewing the evidence on the role of H. pylori infection in extra-gastric diseases is highly important and can shed light on the microbes' roles in health and disease. However, this review lacked a critical assessment of the literature and published studies, and the articles' methodological qualities. The authors should add comments/ section on the quality of available evidence.
While the evidence on the association of H. pylori infection with IDA, vitamin B and PIT is convincing and well established, its role in ophthalmic and dermatological conditions is not convincing or biologically plausible. It is not clear why the authors focused on these conditions and not on metabolic and neurological disorders, allergies, GERD, that were also linked to H. pylori. It is worth adding to the review details on these conditions as well.
The authors need to add a section on possible regarding biological mechanisms and pathways that might explain these associations, including potential confounders and biases.
A section regarding future research directions should be added.
Specific comments
Introduction line 27: please change "contamination" to "colonization" or "infection"
Lines 27-28: H. pylori infection is acquired in early childhood and it is transmitted from mothers/fathers to children and between siblings. The current evidence does not support massive acquisition in adulthood or among sexual partners. Therefore is suggest to revise this sentence.
LINES 47-50: It is important to mention that H. pylori infection mostly causes asymptomatic gastritis, and that peptic disease/ gastric cancer develop only in small fraction of the infected persons.
Author Response
Point by point response to Reviewer 3
We thank the Reviewer for her/his comments
Reviewing the evidence on the role of H. pylori infection in extra-gastric diseases is highly important and can shed light on the microbes' roles in health and disease. However, this review lacked a critical assessment of the literature and published studies, and the articles' methodological qualities. The authors should add comments/ section on the quality of available evidence.
Response: Because of the reviewer’s concern we have now more critically examined the data from the literature and introduced the reasons for discrepancy between studies
While the evidence on the association of H. pylori infection with IDA, vitamin B and PIT is convincing and well established, its role in ophthalmic and dermatological conditions is not convincing or biologically plausible. It is not clear why the authors focused on these conditions and not on metabolic and neurological disorders, allergies, GERD, that were also linked to H. pylori. It is worth adding to the review details on these conditions as well.
Response: We agree with the reviewer’s comment and have therefore added sections on esophageal, metabolic, neurological and allergic disorders
The authors need to add a section on possible regarding biological mechanisms and pathways that might explain these associations, including potential confounders and biases.
Response: We believe that discussing the putative mechanisms underlying the association, be it direct or inverse, between H. pylori infection and a given associated disease, in each specific paragraph is more informative and of easier understanding to the readier rather than having them pooled in a separate paragraph. Also, because of Reviewer 1 suggestion, we have now introduced a table which summarizes the associations between H. pylori and extra-gastric disease highlighting the putative pathogenic meccanism(s). However, should the referee feel that the mechanisms related to the different associations should be pooled in a separate paragraph we have no objection in doing it.
A section regarding future research directions should be added.
Response: Per the reviewer’s request, we have expanded our final conclusion paragraph that is now entitled: conclusion and future research. In this paragraph we have outlined which could be the future investigation in the intriguing field of H. pylori infection and its putative pleiotropic associate conditions.
Specific comments
Introduction line 27: please change "contamination" to "colonization" or "infection"
Response: done
Lines 27-28: H. pylori infection is acquired in early childhood and it is transmitted from mothers/fathers to children and between siblings. The current evidence does not support massive acquisition in adulthood or among sexual partners. Therefore is suggest to revise this sentence
Response: per the reviewer’s request we have made it clear that current evidence does not support massive acquisition in adulthood or among sexual partners and clarified that one study suggests transmission between sexual pattern in a particular setting
LINES 47-50: It is important to mention that H. pylori infection mostly causes asymptomatic gastritis, and that peptic disease/ gastric cancer develop only in small fraction of the infected persons.
Response: This has now been made clear in the revised version of the manuscript.
